# Land inclination controls $CO_2$ and $N_2O$ fluxes, but not $CH_4$ uptake, from a temperate upland forest soil

Lauren M. Gillespie[1], Nathalie Y. Triches[2], Diego Abalos[3], Peter Finke[4], Sophie Zechmeister-Boltenstern[1], Stephan Glatzel[5], Eugenio Díaz-Pinés[1]

[1] Institute of Soil Research, University of Natural Resources and Life Sciences, Vienna (BOKU), Peter-Jordan-Straße 82, 1190, Vienna, Austria.

[2] Department of Environment, Faculty of Bioscience Engineering, Ghent University, Coupure links 653, B-9000, Ghent, Belgium. Present address: Max Planck Research Institute for Biogeochemistry, Hans-Knöll- Straße 10, 07745, Jena, Germany.

[3] Department of Agroecology, iCLIMATE, Aarhus University, Blichers Allé 20, Tjele, 8830, Denmark.

[4] Department of Environment, Research group of soilscape genesis, Ghent University, Coupure links 653, B-9000, Ghent, Belgium.

[5] University of Vienna, Department of Geography and Regional Research, Universitätsstraße 7, 1010 Vienna, Austria.

**Abstract**

Inclination and spatial variability in soil and litter properties influence soil greenhouse gas (GHG) fluxes, and thus on-going climate change, but their relationship in forest ecosystems is poorly understood. To elucidate this, we explored the effect of inclination, distance to a stream, soil moisture, soil temperature, and other soil and litter properties on soil-atmosphere fluxes of carbon dioxide ($CO_2$), methane ($CH_4$), and nitrous oxide ($N_2O$) with automated static chambers in a temperate upland forest in Eastern Austria. We hypothesised that soil $CO_2$ emissions and $CH_4$ uptake are higher in sloped locations with lower soil moisture content, whereas soil $N_2O$ emissions are higher in flat, wetter locations. During the measurement period, soil $CO_2$ emissions were significantly higher on flat locations ($p < 0.05$), and increased with increasing soil temperature ($p < 0.001$) and decreasing soil moisture ($p < 0.001$). The soil acted as a $CH_4$

sink, and $CH_4$ uptake was not significantly related to inclination. However, $CH_4$ uptake was
significantly higher at locations furthest away from the stream compared to at the stream ($p <$
0.001), and positively related to litter weight and soil C content ($p < 0.01$). $N_2O$ fluxes were
significantly higher on flat locations and further away from the stream ($p < 0.05$), and increased
with increasing soil moisture ($p < 0.001$), soil temperature ($p < 0.001$) and litter depth ($p <$
0.05). Overall, this study underlines the importance of inclination and the resulting soil and
litter properties in predicting GHG fluxes from forest soils and therefore their potential source-
sink balance.

**Keywords**: slope inclination, soil greenhouse gas fluxes, carbon dioxide, methane, nitrous
oxide, soil moisture, forest litter

**Introduction**
Forests play a crucial role in the global climate by emitting and consuming the greenhouse gases
(GHGs) carbon dioxide ($CO_2$), methane ($CH_4$), and nitrous oxide ($N_2O$) (IPCC, 2022). They
store a large amount of carbon (C) in the vegetation and soil organic matter and can be effective
$CO_2$ sinks (Pan et al., 2011). Soil microorganisms also take up atmospheric C through the
oxidisation of $CH_4$ during methanotrophy (Le Mer and Roger, 2001; Hiltbrunner et al., 2012).
However, forest soils also emit substantial quantities of $CO_2$ (Webster et al., 2008), which, in
aerobic conditions, is mainly released by root respiration and microbial respiration during
decomposition (Cronan, 2018; Zechmeister-Boltenstern et al., 2018). $N_2O$ is produced by soil
microorganisms, mainly during nitrification and denitrification (Butterbach-Bahl et al., 2013).
In aerobic conditions, bacteria convert ammonium to nitrite and further to nitrate during
nitrification. In anoxic conditions, nitrate is then used as an alternative electron acceptor instead
of $O_2$ and reduced to $N_2$ during denitrification (Butterbach-Bahl et al., 2014). Under most
conditions, these processes occur simultaneously and usually result in a net atmospheric
emission of $N_2O$ (Ambus, 1998). Conversely, net $N_2O$ uptake has been reported from forest
soils, especially since monitoring instrumentation has become sensitive enough to measure very
low fluxes (Savage et al., 2014; Subke et al., 2021). Net $N_2O$ uptake (from the atmosphere into
the soil) is a complex process closely tied to $N_2O$ consumption (within the soil) that is driven
principally by denitrifying bacteria (Liu et al., 2022).
Temporal and spatial variations in soil $CO_2$, $CH_4$, and $N_2O$ fluxes are driven mostly by changes
in soil temperature and soil moisture (Raich and Potter, 1995; Davidson et al., 1998; Le Mer
and Roger, 2001; Butterbach-Bahl et al., 2014). Rising temperatures accelerate microbial
activities and, consequently, the production and emission of $N_2O$ and $CO_2$ (Butterbach-Bahl et
al., 2013). Elevated soil respiration could lead to a depletion of $O_2$, which also results in
increased $N_2O$ from denitrification (Butterbach-Bahl et al., 2013). Contrarily, $CH_4$ uptake
appears to be less sensitive to temperature changes than $CO_2$ and $N_2O$ fluxes (Hanson and
Hanson, 1996). Soil moisture has a major influence on all GHG fluxes by regulating $O_2$ and
substrate availability to soil microorganisms and influencing the diffusion of gases within the
soil matrix (Butterbach-Bahl et al., 2014; Schimel, 2018). Indeed, soil microbial activity
decreases as soils become water saturated (Davidson et al., 2012). Soil moisture further affects
fluxes since diffusion coefficients of GHG in air are approximately $10^4$ times larger than in
water (Marrero and Mason, 1972).
Inclination and distance to a water source influence some of the most important drivers of soil
GHG fluxes. For example, soil moisture content changes on small scales at different
inclinations through accumulation, runoff, and leaching of precipitation water (Burt and
Butcher, 1985; Lookingbill and Urban, 2004; Lin et al., 2006). Inclination also modifies other
important drivers of soil GHG fluxes, such as the hydrological transport of nutrients (Hairston
and Grigal, 1994), litter accumulation (Butler et al., 1986), soil aeration, soil texture, soil pH,
and substrate availability (soil C and N), usually resulting in a high GHG spatial variability
(e.g., Fierer and Jackson, 2006; Thomas and Packham, 2007). Flat locations by a water source
are also at higher risk to be influenced by flooding and subsequent changes to the soil properties
and soil microbial community (Ou et al., 2019; Unger et al., 2009). Forest litter in particular
can have a major impact on the exchange of GHGs by adding nutrients to the soil, acting as a
physical barrier (i.e., holding gases in the soil rather than releasing them into the atmosphere)
or influencing the water and heat exchange between soil and atmosphere (Leitner et al., 2016;
Walkiewicz et al., 2021).
Studies on the effect of inclination on GHG fluxes from temperate upland forest soils are
particularly rare. Some studies reported higher soil $CO_2$ emissions on sloped compared to flat
locations, associated with warmer air and soil temperatures and lower soil moisture contents,
favouring faster diffusion rates though not so low as to impede microbial activity (Yu et al.,
2008; Warner et al., 2018). Conversely, no effect of topography on soil $CO_2$ emissions has also
been reported in a laboratory study from a montane tropical forest (Arias-Navarro et al., 2017).
With regard to $CH_4$, relatively little is known on how inclination and its influence on chemical
and physical soil properties may affect $CH_4$ fluxes (Warner et al., 2018). Soil $CH_4$ uptake is
highly variable in space and time and appears to be highest on dry slopes (Hiltbrunner et al.,
2012; Yu et al., 2021), even though it is assumed that temperate upland forest soils take up $CH_4$
irrespective of the inclination (Lamprea Pineda et al., 2021). Effects of inclination on $N_2O$
fluxes are also contradictory. Some studies show increased $N_2O$ emissions with higher soil
water content at flat locations (Davidson et al., 2000; Lamprea Pineda et al., 2021), whereas
others show a higher emission in aerated soils on slopes (Yu et al., 2008, 2021). Assessing the
impact of inclination on soil GHGs therefore remains a challenging task.
In this study, we aim to improve the understanding of the effects of inclination and distance to
a stream on the emission and uptake of GHGs in a temperate upland forest soil in Eastern
Austria. We monitored soil $CO_2$, $CH_4$, and $N_2O$ fluxes with automated chambers over six
months for two different inclinations and at four distances from a stream in a deciduous forest.
We tested three hypotheses: 1) Soil $CO_2$ emissions are higher in sloped than flat locations
because of the inclination and the lower soil moisture content at sloped locations; 2) Soil $CH_4$
uptake is higher in sloped than flat locations because of the inclination and the lower soil
moisture content at sloped locations; and 3) Soil $N_2O$ emissions are lower in sloped than flat
locations because of the inclination and the higher soil moisture content at flat locations.

**Methods**
*Study site and experimental design*
This study was conducted within the framework of the "Long-Term Ecosystem and socio-
ecological Research Infrastructure - Carbon, Water and Nitrogen" (LTER-CWN) project
(further information is available at https://www.lter-austria.at/en/cwn-sites/). The BOKU
University Forest "Rosalia Lehrforst" (47°42'25.35" N / 16°16'36.62" E) is one of the
associated sites and served as the site for our study (see Fürst et al., 2021) for more information).
At the site, European beech (*Fagus sylvatica* L.) and Norway spruce (*Picea abies* (L.) H. Karst.)
are the dominant tree species, but alluvial forest species (*Alnus spp.* Mill*, Fraxinus excelsior*
L.) are also present next to the study location. The elevation is around 400 m a.s.l. and the
dominant soil type is pseudo-gleyic Cambisol (Schad, 2016).
We used the GasFluxTrailer (explained below) to measure soil GHG fluxes from 17 June to 24
November 2020. We positioned 16 chambers linearly in groups of four at four different
distances from a small forest stream: 0.5 m, 5 m, 10 m, and 15 m (Fig. S1). Adjacent trees to
the chambers were *F. sylvatica* and *P. abies*. These distances served as first treatment effect
and are hereafter referred to as chamber group (CG): CG0.5, CG5, CG10, and CG15. These
distances were chosen because they were expected to cover a decreasing soil moisture gradient
from CG0.5 towards CG15. To measure this gradient, one Em50 (METER Group, Inc. Pullman,
WA; USA) was connected to four ECH2O 5 TM volumetric water content and temperature
sensors (METER Group). One sensor was installed per CG approximately one meter away from
the outer chamber (Fig S.1). As a second treatment effect, the distances were also chosen so

that the CGs were set up at two different inclinations. CG0.5 and CG5 were located at flat (average 1°; the slope at these distances did not exceed 2°), CG10 and CG15 at sloped locations (average 35°; west-facing).

For meteorological information, we used the precipitation (OTT Pluvio L weighing rain gauge) and air temperature (air temperature and humidity sensor TR1) data recorded at 30 min intervals by the weather station "Mehlbeerleiten", located approximately 100 m north-west of the site (Diaz-Pines and Gasch, 2021; Fürst et al., 2021).

*Gas flux measurements: GasFluxTrailer*

An automated and mobile measuring system was used, termed the GasFluxTrailer. It consists of a mobile trailer estimating soil-atmosphere GHG exchange rates of $CH_4$, $CO_2$, and $N_2O$. The GasFluxTrailer connects with the chambers, and it controls the sampling of each individual chamber (i.e., the opening and closing and gas sampling) and recording of the gas concentrations. The 16 automated, static, non-steady-state, non-flow-through chambers (Pumpanen et al., 2004) with an area of 0.5 m × 0.5 m and height of 0.15 m are made of stainless-steel and placed on stainless-steel frames of the same area. They are equipped with fans to ensure homogenous air mixing. The gas analysers are a G2301 (PICARRO Inc., Santa Clara, USA), measuring concentrations of $CO_2$ and $CH_4$, and a G5131i (PICARRO Inc.), estimating $N_2O$ concentrations. The software used to run automatic sequences is the IDASw Recorder 4.5.0., developed by the Institute of Meteorology and Climate Research Atmospheric Environmental Research (IMK-IFU) in Germany.

*Field and laboratory measurements*

We inserted the chamber frames 5 cm deep into the soil approximately one month before the measuring campaign to avoid additional soil $CO_2$ release from cut roots, affecting our measurements (Davidson et al., 2002). For each measurement estimate, a chamber was closed

for 10 min, which, thanks to the highly sensitive instruments used here, was sufficient time to
measure gas concentrations changes, including low $N_2O$ fluxes (Harris et al., 2021). The closing
and opening was done successively; thus one full cycle of all 16 chambers took 160 minutes.
We calculated fluxes with a linear regression approach according to Butterbach-Bahl et al.
(2011). This was justified with short chamber closure times and a relatively large chamber size
(Hutchinson and Mosier, 1981). Positive flux values indicate gas emission from the soil, and
negative values indicate net uptake. To ensure the system was running and working correctly,
we controlled the GHG flux measurements on-site every week and three-four times a week
remotely. There were no inundations or significant drying/rewetting events during the
observation period.
Close to each of the 16 chambers, a litter and soil sample was collected in December 2021. The
litter depth was measured first, before disturbing the litter and topsoil by placing a 0.2 m × 0.2
m frame on the ground at this location. The litter was then collected within this frame, dried at
65°C for 7 days, and weighed. After litter collection and removal of organic layer, two soil
cores (stainless steel core, 7 cm diameter, 7 cm depth) were taken from the topsoil mineral layer
for analyses of pH, C and N content, and soil texture. C and N contents (%) were determined
by dry combustion on 1.6 mg of soil using the Austrian standard ÖNORM L 1080 (ÖNORM,
2013). Particle size analysis was conducted using the pipette method on 10 g of soil according
to the Austrian standard ÖNORM L 1061 (ÖNORM, 2002), after the organic material had been
burned off in an oven at 550 °C, to determine soil texture (%). In short, sieved soil (<2 mm) is
agitated in a volume of water, and a pipette is used to sample a defined volume at a defined
depth at specific times after which the samples are dried to determine clay and silt contents.
The remaining soil is then sieved (63 µm) to determine sand content. Soil pH was measured on
5 g of soil with 0.01 $M$ $CaCl_2$ using the Austrian standard ÖNORM L 1083 (ÖNORM, 2006).
Because the soil was relatively rocky, we calculated the soil bulk density (BD, $g\ cm^{-3}$) including
the coarse (stone) fraction as:
$$BD \ with \ stones = \frac{dry \ soil \ weight}{core \ volume}$$
where dry soil weight is the weight of the soil in the core after oven drying in g and core volume
is the volume of the core in $cm^3$.
We calculated the total porosity ($\Phi$) using the bulk density and an estimated soil particle density,
obtained by a weighted average of the specific weights of mineral material (2.65 g $cm^{-3}$) and
organic matter (1.45 g $cm^{-3}$). We took into account the organic matter content because it was
relatively high, i.e., between 8 and 27 %.

*Data processing and statistics*
We quality-controlled the $CO_2$, $CH_4$, and $N_2O$ flux data using the determination coefficient (R-
squared, $R^2$) values between GHG concentrations and the time after chamber closure. For $CO_2$
and $CH_4$, we filtered the data with $R^2 > 0.8$ and a visual plausibility check based on expert
knowledge. For $N_2O$, $R^2 > 0.8$ was applied only if fluxes were $> 5$ µg $N_2O$-N $m^{-2}$ $h^{-1}$. For low
flux rates ($< 5$ µg $N_2O$-N $m^{-2}$ $h^{-1}$), we did not remove values with $R^2 < 0.8$ if corresponding $CO_2$
fluxes were valid. We kept these measurements in the dataset, because the low $R^2$ values were
due to fluxes below the detection limit of the system; however, the measurement itself remained
valid as indicated by plausible $CO_2$ fluxes, and as elaborated in Parkin et al. (2012). Through
this quality control, we found that two chambers did not produce any reliable measurements
from 24 September onwards. August data for all chambers was excluded due to malfunctioning
of the equipment that was not initially detected. Furthermore, all the data from one chamber
(chamber 13) were also not used for the analysis because of a failure in the chamber gas
sampling. After data quality screening, there were 125 measurement days included in analysis
for $CO_2$ and $CH_4$, and 85 days for $N_2O$.
All statistical analyses were performed with R (version 4.0.4; R Core Team, 2022). All data
was visually and statistically checked for normality (Levene's test) and homoscedasticity before

testing for statistical differences. Since the original data was not normally distributed, $CO_2$ and $N_2O$ fluxes were log-transformed. To homogenise the data from the gas flux analysers and the soil temperature and soil moisture sensors, we rounded all gas flux data to 3-hour intervals (00:00, 03:00, 06:00, 09:00, 12:00, 15:00, 18:00, 21:00), corresponding to the approximate gas flux measurement cycle duration. Soil temperature and soil moisture data was available every 30 min and was thus also aggregated for the same 3-hour intervals. For the statistical analyses, we ran linear mixed-effect models (LMM) using the "lmer" function from the lme4 package (version 1.1-27; Bates et al., 2015), the "lmerTest" package (version 3.1-3; Kuznetsova et al., 2017), and the "optimx" function from the optimx package (version 2021-6.12; Nash and Varadhan, 2011). Models were selected according to the guidelines of Zuur et al. (2009). For the null models, soil temperature, soil moisture, and inclination or distance from the stream (i.e., 0.5 m to 15 m away from the stream, CG0.5 – CG15) were included as fixed effects, with an interaction between soil temperature and soil moisture. Sampling date and chamber number were included as random effects. Sampling date was included as a random variable since we were not exploring temporal changes and since there were multiple observations per day. Inclination and distance were not included in the same model because they were highly correlated. We therefore separated our treatments in "inclination" and "distance", resulting in two LMM models per GHG. We then created a model, using the original model structure, including each soil or litter characteristic individually as an additional explanatory variable. The model Akaike Information Criterions (AIC) were then compared using ANOVA. Finally, we selected the model with the lowest AIC value if it was significantly different from the null model. This was done for each gas-inclination or distance combination. To obtain the conditional and marginal $R^2$ of the models, the "r2_nakagawa" function from the performance package was used (version 0.7.3; Nakagawa et al., 2017).

**Results**

Over the measurement period (June-November 2020, 161 days), the mean air temperature was
12.30°C and cumulative precipitation was 561 mm. The average volumetric water content, here
referred to as 'soil moisture', was $0.22 \pm 0.07 \, \text{m}^3 \, \text{m}^{-3}$, with wetter soils in flat
$(0.28 \pm 0.04 \, \text{m}^3 \, \text{m}^{-3})$ compared to sloped locations $(0.17 \pm 0.02 \, \text{m}^3 \, \text{m}^{-3}$; Fig. S2). The mean soil
temperature was $12.85 \pm 2.62$°C, with no significant difference between flat and sloped
locations. The wettest and warmest location was at CG5 $(0.31 \pm 0.03 \, \text{m}^3 \, \text{m}^{-3}$ and $13.62 \pm 2.54$°C;
Fig. S2). Changes in soil moisture and soil temperature were strongly related to variation of
precipitation and air temperature (Fig. S3). Furthermore, the interaction between soil moisture
and soil temperature was significant in all models ($p < 0.001$), showing a decrease in soil
moisture with increasing soil temperature. Litter depth and weight were much lower at CG0.5
than at all other CGs (Table 1). Soil N and C contents and organic matter content were lowest
at CG0.5 and highest at CG10, but C:N ratios were similar at all CGs (Table 1). Bulk density
was low (0.6-0.8 g cm$^{-3}$) at all distances. Soil pH was considerably higher at CG0.5 compared
to all other CGs (Table 1). The soil in flat locations was sandier, whereas the sloped locations
were more clayey (Table 1).
**Table 1:** Average value and standard error of litter and soil parameters at each distance from
the stream. "CG" indicates chamber group, with the numbers 0.5, 5, 10, and 15 defining the
distance to the stream (m). Different letters indicate differences between distances (Dunn
multiple comparison test after Kruskal–Wallis test, $p < 0.05$) for each variable.

| Variable | Unit | Chamber group | | | |
|---|---|---|---|---|---|
| | | CG0.5 | CG5 | CG10 | CG15 |
| Litter depth | cm | $4.4 \pm 0.7^a$ | $7.0 \pm 1.2^{ab}$ | $8.5 \pm 1.0^b$ | $8.0 \pm 1.4^b$ |
| Litter weight | g m$^{-2}$ | $147.7 \pm 23.1^a$ | $311.8 \pm 47.0^{ab}$ | $358.5 \pm 100.0^{ab}$ | $622.2 \pm 362.1^b$ |
| Soil N content | % | $0.25 \pm 0.06^a$ | $0.39 \pm 0.09^{ab}$ | $0.6 \pm 0.26^b$ | $0.42 \pm 0.18^{ab}$ |
| Soil C content | % | $4.12 \pm 0.78^a$ | $6.35 \pm 1.65^{ab}$ | $10.15 \pm 4.8^b$ | $7.85 \pm 4.29^{ab}$ |
| Soil C:N ratio | - | $16.56 \pm 1.35^a$ | $16.24 \pm 0.81^a$ | $17.07 \pm 1.81^a$ | $18.23 \pm 1.99^a$ |
| Bulk density* | g cm$^{-3}$ | $0.81 \pm 0.15^a$ | $0.73 \pm 0.12^a$ | $0.6 \pm 0.11^a$ | $0.81 \pm 0.08^a$ |
| Volumetric stone content | % | $7.59 \pm 8.4^a$ | $7.84 \pm 2.57^a$ | $10.79 \pm 2.78^a$ | $13.16 \pm 2.24^a$ |
| Porosity† | - | $0.75 \pm 0.01^a$ | $0.79 \pm 0.03^{ab}$ | $0.87 \pm 0.04^b$ | $0.80 \pm 0.02^{ab}$ |
| Organic material (OM) | % | $9.25 \pm 1.4^a$ | $13.87 \pm 3.73^{ab}$ | $20.86 \pm 8.01^b$ | $16.70 \pm 7.02^{ab}$ |
| Soil pH | - | $5.57 \pm 0.65^a$ | $4.00 \pm 0.34^{ab}$ | $4.01 \pm 0.34^{ab}$ | $3.78 \pm 0.31^b$ |

| | | | | | |
|---|---|---|---|---|---|
| Sand content | % | 598.970 ± 7.5[a] | 52.0 ± 9.5[a] | 40.6 ± 3.7[a] | 41.6 ± 4.4[a] |
| Silt content | % | 38.5 ± 7.7[a] | 45.1 ± 8.5[a] | 53.1 ± 4.5[a] | 52.0 ± 5.0[a] |
| Clay content | % | 2.5 ± 0.3[a] | 2.9 ± 1.4[a] | 6.3 ± 1.4[b] | 6.5 ± 0.7[ab] |

*with coarse material
†without coarse material


*Soil $CO_2$ emissions*
The average soil $CO_2$ emissions during the observation period were
$116.2 \pm 61.5$ mg $CO_2$-C m$^{-2}$ h$^{-1}$, with flat and sloped locations emitting $113.6 \pm 66.7$ and
$118.6 \pm 56.3$ mg $CO_2$-C m$^{-2}$ h$^{-1}$, respectively (Table 2, Fig. 1a). The soil $CO_2$ emission pattern
was bell-curved with increasing distance from the stream, with the lowest emissions at CG0.5,
the highest emissions at CG5 and CG10, and relatively low emissions at CG15 as compared to
CG10 (Table 2, Fig. 1a). Our analysis showed a significant inclination effect on soil $CO_2$
emissions ($p < 0.05$); furthermore, we found a significant difference between emissions at
CG0.5 and CG5 ($p < 0.001$), as well as between CG0.5 and CG10 ($p < 0.05$, Table 2).

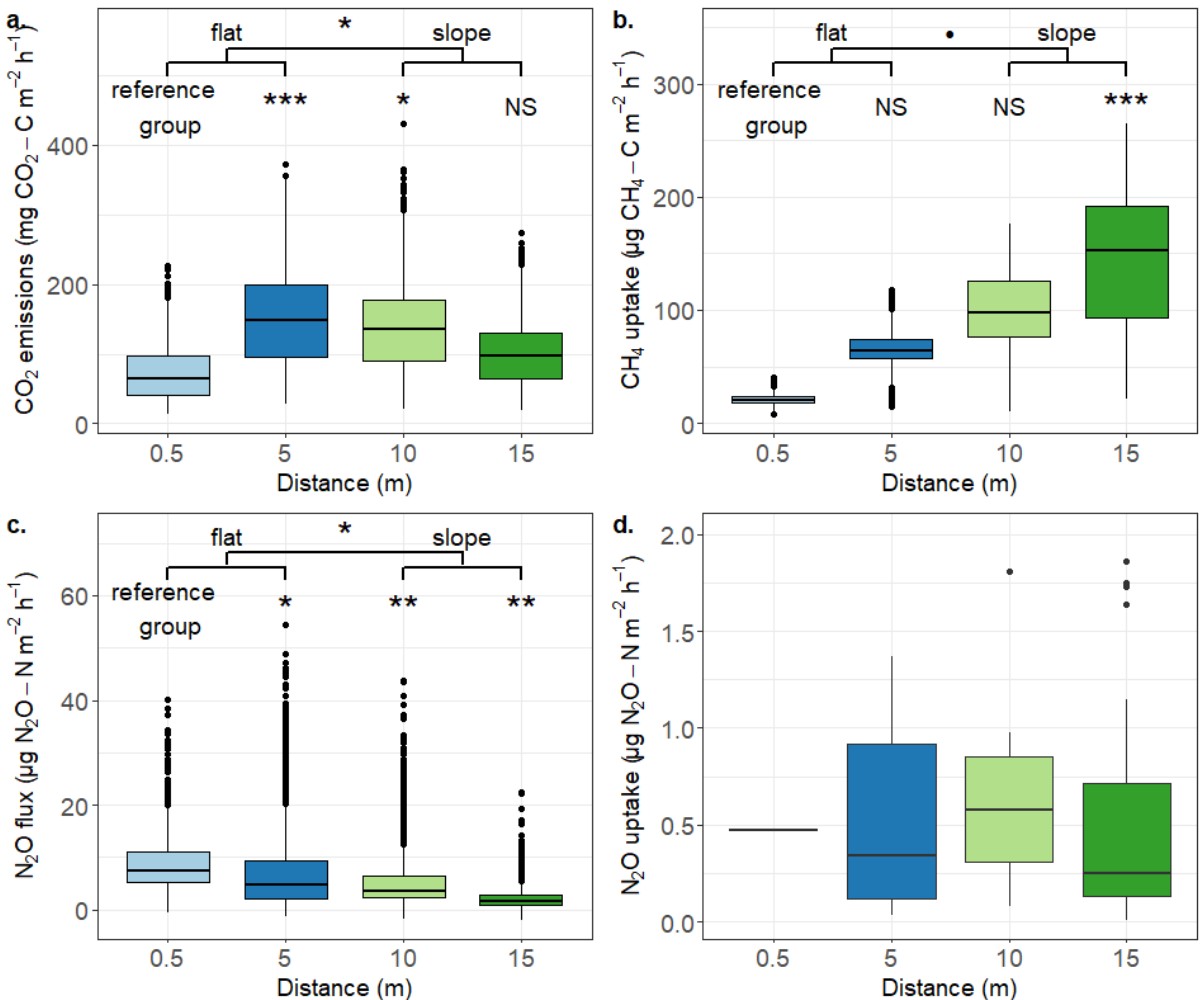

Figure 1: **a.** $CO_2$ emissions (mg $CO_2$-C $m^{-2}$ $h^{-1}$), **b.** $CH_4$ uptake ($\mu$g $CH_4$-C $m^{-2}$ $h^{-1}$), **c.** $N_2O$ flux ($\mu$g $N_2O$-N $m^{-2}$ $h^{-1}$), and **d.** $N_2O$ uptake ($\mu$g $N_2O$-N $m^{-2}$ $h^{-1}$) at four distances from a stream: 0.5 m, 5 m, 10 m, and 15 m (i.e., Chamber Groups: CG0.5, CG5, CG10, and CG15). Blue indicates flat locations, and green indicates sloped locations. Statistical significances are from the 'distance model' (linear mixed model, LMM) for the differences between the four distances and the 'inclination model' for the differences between the flat and slope positions associated with each gas (Table 1, 2, 3); no LMM was run for $N_2O$ uptake. Non-significance is indicated by 'NS' and *p*-values are coded as $p < 0.1$ '.', $p < 0.05$ '*', $p < 0.01$ '**', and $p < 0.001$ '***'.


**Table 2:** LMM results exploring the relationship between inclination (flat compared to slope)
or distance (m), soil moisture ($m^3 m^{-3}$), soil temperature (°C), soil moisture:soil temperature
interaction, soil pH, and volumetric stone content on soil $CO_2$ emissions (mg $CO_2$-C $m^{-2} h^{-1}$).
Soil pH and volumetric stone content are included because the LMM models including these
variables had AIC values statistically smaller than the null model. $R^2m$ indicates marginal $R^2$,
and $R^2c$ indicates conditional $R^2$ values. *P*-values are coded as: $p < 0.05$ '*', $p < 0.01$ '**', and
$p < 0.001$ '***'.

| $CO_2$ emissions | $R^2c=$ 0.91 | | | $R^2m=$ 0.28 | AIC= | -9475.99 |
|---|---|---|---|---|---|---|
| **Inclination – pH** | Estimate | Std. Error | df | t value | Pr(>\|t\|) | |
| Soil moisture | -1.48 | 0.18 | 11330.00 | -8.22 | < 2E-16 | *** |
| Soil temperature | 0.06 | 4.55E-03 | 9060.00 | 14.08 | < 2E-16 | *** |
| Inclination (slope) | -0.41 | 0.17 | 12.20 | -2.42 | 0.03 | * |
| Moisture:temperature | -0.05 | 0.01 | 11410.00 | -4.35 | 1.40E-05 | *** |
| Soil pH | -0.41 | 0.12 | 12.00 | -3.33 | 6.02E-03 | ** |
| $CO_2$ emissions | $R^2c=$ 0.91 | | | $R^2m=$ 0.42 | AIC= | -9474.05 |
| **Distance – stone content** | Estimate | Std. Error | df | t value | Pr(>\|t\|) | |
| Soil moisture | -1.49 | 0.18 | 11300.00 | -8.26 | < 2E-16 | *** |
| Soil temperature | 0.06 | 4.55E-03 | 9060.00 | 14.07 | < 2E-16 | *** |
| Distance 5 m | 0.86 | 0.16 | 10.10 | 5.52 | 2.49E-04 | *** |
| Distance 10 m | 0.43 | 0.16 | 10.10 | 2.76 | 0.02 | * |
| Distance 15 m | 0.14 | 0.16 | 10.10 | 0.86 | 0.41 | |
| Moisture:temperature | -0.05 | 0.01 | 11400.00 | -4.35 | 1.39E-05 | *** |
| Volumetric stone content | 0.02 | 0.01 | 10.00 | 1.76 | 0.11 | |


Both model results showed a significant negative correlation between soil $CO_2$ emissions and
soil moisture ($p < 0.001$, Table 2). This pattern was more distinct looking at the CGs at the
different distances (Fig. 2a). A significant positive correlation between $CO_2$ emissions and soil
temperature was found ($p < 0.001$, Table 2, Fig. 2b). The interaction between soil moisture and
temperature, namely soil moisture decreasing with increasing soil temperature, was shown to
correlate negatively with $CO_2$ emissions ($p < 0.001$, Table 2). According to "inclination" model
results, $CO_2$ emissions also decreased with increasing soil pH when comparing flat to sloped
locations ($p < 0.01$, Table 2).

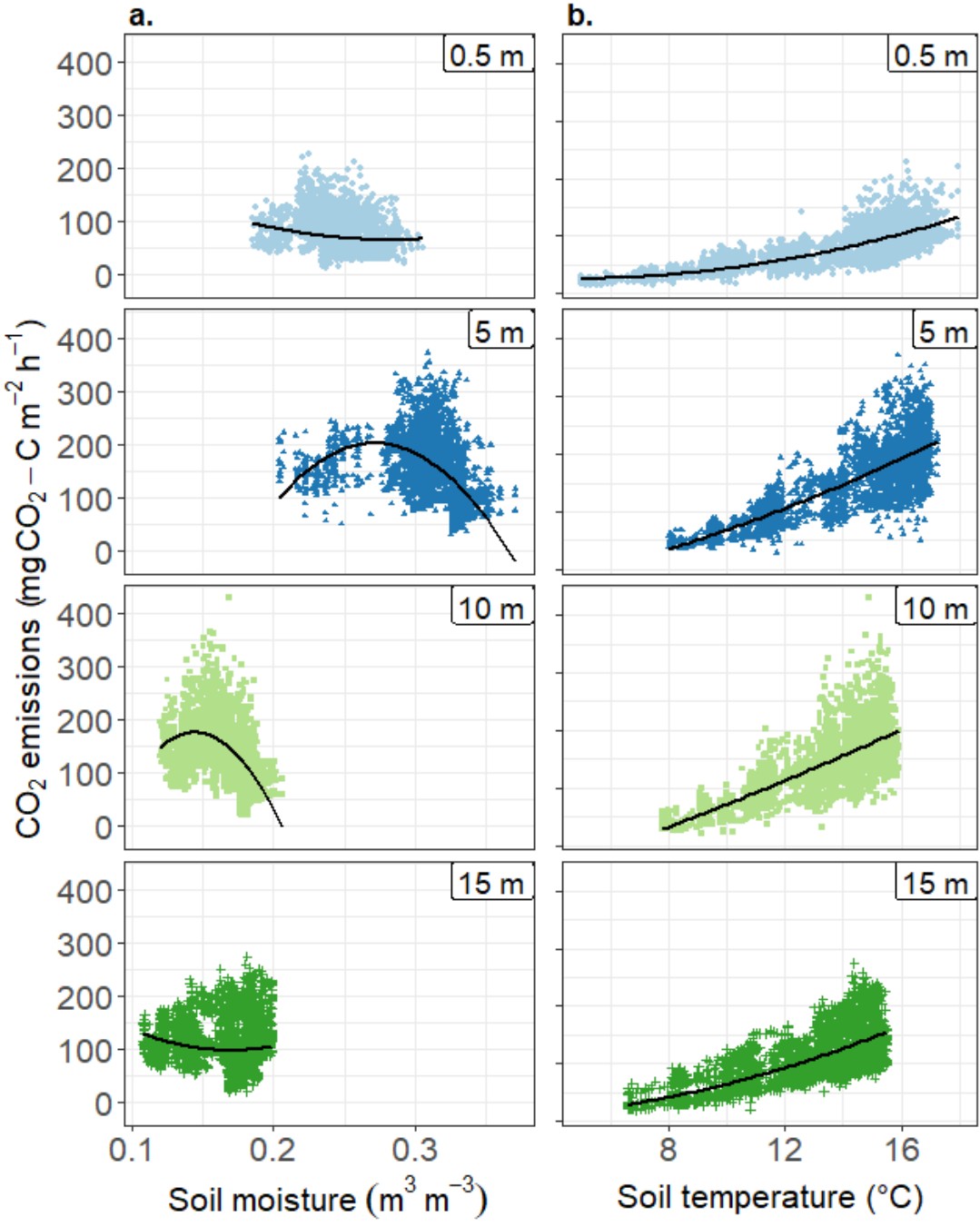

**Figure 2:** Relationship between soil $CO_2$ emissions (mg $CO_2$-C m$^{-2}$ h$^{-1}$) and **a**. soil moisture (m$^3$ m$^{-3}$), and **b**. soil temperature (°C) by distance from the stream (0.5 m, 5 m, 10 m, 15 m). Flat locations are indicated in blue (0.5 m and 5 m) and sloped locations in green (10 m and 15 m). The fitted lines show the linear regression on geometrically distributed data using the "geom_smooth" function (method = "lm") from ggplot2. The $R^2$ for these regressions are shown in Table 2.

297

298

*Soil CH$_4$ uptake*

The soil showed an average CH$_4$ uptake of $88.5 \pm 58.0$ µg CH$_4$-C m$^{-2}$ h$^{-1}$, with uptake 180 % higher in sloped as compared to flat locations ($126.9 \pm 51.3$ and $45.0 \pm 25.3$ µg CH$_4$-C m$^{-2}$ h$^{-1}$, respectively; Fig. 1b). Average CH$_4$ uptake increased by approximately 40 µg CH$_4$-C m$^{-2}$ h$^{-1}$ per 5 m distance further away from the stream (Fig. 1b). However, the "inclination" model showed only marginally significant differences between the CH$_4$ uptake at flat and sloped locations ($p < 0.1$, Table 3). Litter weight was positively correlated with the CH$_4$ uptake at flat and sloped locations ($p < 0.001$). The "distance" model showed a significant difference between the locations at the stream (CG0.5) and furthest away (CG15; $p < 0.001$, Table 3) and a positive correlation between soil C content and CH$_4$ uptake at all CGs ($p < 0.01$, Table 3).

**Table 3:** LMM results exploring the relationship between inclination (flat compared to slope) or distance (m), soil moisture (m$^3$ m$^{-3}$), soil temperature (°C), soil moisture:soil temperature interaction, litter weight (g), and soil C content effects on soil CH$_4$ uptake (µg CH$_4$-C m$^{-2}$ h$^{-1}$). Litter weight and soil C content are included because the LMM models including these variables had AIC values statistically smaller than the null model. R$^2$m indicates marginal R$^2$, and R$^2$c indicates conditional R$^2$ values. *P*-values are coded as: $p < 0.1$ '.', $p < 0.05$ '*', $p < 0.01$ '**', and $p < 0.001$ '***'.

| CH$_4$ uptake | R$^2$c= 0.97 | | R$^2$m= 0.67 | | AIC= 88007.79 | |
|---|---|---|---|---|---|---|
| **Inclination –** | | | | | | |
| **Litter weight** | Estimate | Std. Error | df | t value | Pr(>|t|) | |
| Soil moisture | 173.06 | 12.81 | 11318.95 | 13.51 | < 2E-16 | *** |
| Soil temperature | -2.52 | 0.33 | 10140.69 | -7.71 | 1.43E-14 | *** |
| Inclination (slope) | 30.51 | 15.49 | 12.11 | 1.97 | 0.07 | . |
| Moisture:temperature | -14.73 | 0.80 | 11406.27 | -18.34 | < 2E-16 | *** |
| Litter weight | 0.80 | 0.16 | 12.00 | 4.92 | 3.54E-4 | *** |
| **CH$_4$ uptake** | R$^2$c= 0.97 | | R$^2$m= 0.70 | | AIC= 87987.56 | |

| Distance – Soil C content | Estimate | Std. Error | df | t value | Pr(>|t|) | |
|---|---|---|---|---|---|---|
| Soil moisture | 172.71 | 12.81 | 11313.21 | 13.48 | < 2E-16 | *** |
| Soil temperature | -2.52 | 0.33 | 10139.66 | -7.71 | 1.41E-14 | *** |
| Distance 5 m | 31.93 | 18.74 | 10.02 | 1.70 | 0.12 | |
| Distance 10 m | 24.10 | 22.20 | 10.02 | 1.09 | 0.30 | |
| Distance 15 m | 93.49 | 19.82 | 10.02 | 4.72 | 8.15E-04 | *** |
| Moisture:temperature | -14.73 | 0.80 | 11406.02 | -18.34 | < 2E-16 | *** |
| Soil C content | 7.82 | 2.04 | 10.00 | 3.83 | 3.3E-03 | ** |


Both "inclination" and "distance" model results show a significant, positive correlation between
soil moisture and $CH_4$ uptake ($p < 0.001$), and a significant, negative correlation between soil
temperature and $CH_4$ uptake ($p < 0.001$, Table 3). These patterns could, however, not be
confirmed visually (Fig. 3). Like for $CO_2$ emissions, the soil moisture:soil temperature
interaction, namely soil moisture decreasing with increasing soil temperature, was significant
($p < 0.001$, Table 3). According to the "inclination" model results, litter weight was positively
correlated with the $CH_4$ uptake at flat and sloped locations ($p < 0.001$). The "distance" model
showed that higher soil C content resulted in a higher $CH_4$ uptake at all CGs ($p < 0.01$, Table

326 3).

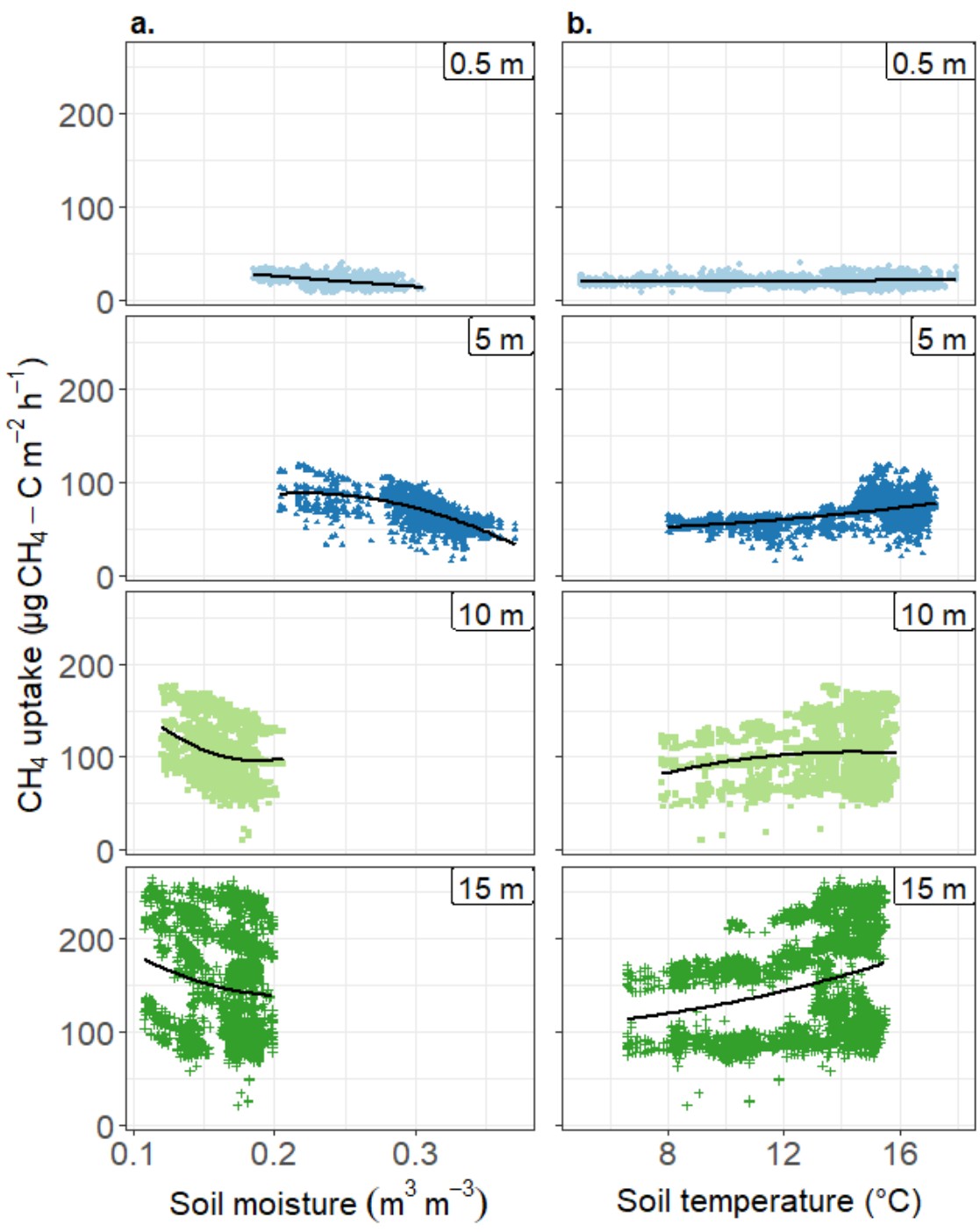

**Figure 3:** Relationship between $CH_4$ uptake ($\mu g$ $CH_4$-C $m^{-2}$ $h^{-1}$) and **a**. soil moisture ($m^3$ $m^{-3}$), and **b**. soil temperature (°C) by distance from the stream (0.5 m, 5 m, 10 m, 15 m). Flat locations are indicated in blue (0.5 m and 5 m) and sloped locations in green (10 m and 15 m). The fitted lines show the linear regression on geometrically distributed data using the "geom_smooth" function (method = "lm") from ggplot2. The $R^2$ for these regressions are shown in Table 3.

 *Soil N$_2$O flux*

The soil had an average N$_2$O emission of $5.9 \pm 6.3$ µg N$_2$O-N m$^{-2}$ h$^{-1}$, with flat locations having

120% higher fluxes than sloped ($8.4 \pm 7.2$ and $3.8 \pm 4.5$ µg N$_2$O-N m$^{-2}$ h$^{-1}$, respectively;

Fig. 1c). The "inclination" model results showed significantly decreasing N$_2$O emissions on

sloped locations compared to flat locations ($p < 0.05$, Table 3). This was supported by the

"distance" model results, with significantly decreasing emissions from CG0.5 towards CG15

(Fig. 1c, Table 4).

**Table 4:** LMM results exploring the relationship between inclination (flat compared to sloped)
or distance (m), soil moisture (m$^3$ m$^{-3}$), soil temperature (°C), soil moisture:soil temperature
interaction, and litter depth (cm) on soil N$_2$O emissions (µg N$_2$O-N m$^{-2}$ h$^{-1}$). Litter depth is
included because the LMM model including this variable had an AIC value statistically smaller
than the null model. R$^2$m indicates marginal R$^2$, and R$^2$c indicates conditional R$^2$ values. *P*-
values are coded as: $p < 0.05$ '*', $p < 0.01$ '**', and $p < 0.001$ '***'.

| N$_2$O emissions | R$^2$c= | 0.79 | | R$^2$m= | 0.21 | AIC= | 4993.94 |
|---|---|---|---|---|---|---|---|
| **Inclination** | Estimate | Std. Error | df | t value | Pr(>|t|) | | |
| Soil moisture | 7.75 | 0.62 | 7660.60 | 12.46 | < 2E-16 | *** | |
| Soil temperature | 0.16 | 0.01 | 3119.98 | 11.42 | < 2E-16 | *** | |
| Inclination (slope) | -0.62 | 0.23 | 13.61 | -2.71 | 0.02 | * | |
| Moisture:temperature | -0.58 | 0.04 | 7445.77 | -14.07 | < 2E-16 | *** | |
| N$_2$O emissions | R$^2$c= | 0.80 | | R$^2$m= | 0.39 | AIC= | 4995.59 |
| **Distance – Litter depth** | Estimate | Std. Error | df | t value | Pr(>|t|) | | |
| Soil moisture | 7.74 | 0.62 | 7650.00 | 12.45 | < 2E-16 | *** | |
| Soil temperature | 0.16 | 0.01 | 3120.00 | 11.40 | < 2E-16 | *** | |
| Distance 5 m | -0.82 | 0.35 | 10.10 | -2.35 | 0.04 | * | |
| Distance 10 m | -1.51 | 0.45 | 10.00 | -3.36 | 7.24E-03 | ** | |
| Distance 15 m | -1.81 | 0.42 | 10.10 | -4.36 | 1.42E-03 | ** | |
| Moisture:temperature | -0.58 | 0.04 | 7440.00 | -14.04 | < 2E-16 | *** | |
| Litter depth | 0.25 | 0.09 | 9.99 | 2.70 | 0.02 | * | |

We found significant positive correlations between N$_2$O emissions and both soil moisture and

soil temperature in both the "inclination" and "distance" model ($p < 0.001$, Table 3). The

correlation between N$_2$O emissions and soil moisture appeared bell-curved at CG5 and CG10

(Fig. 4a). The correlation between $N_2O$ emissions and soil temperature appeared bell-curved at
CG10 (Fig. 4b). As for $CO_2$ and $CH_4$ fluxes, the soil moisture:soil temperature interaction
resulted in significantly decreasing $N_2O$ emissions across all CGs and both the flat and sloped
locations. Similar to the "inclination" model results for $CH_4$ uptake, the $N_2O$ "distance" model
showed that a higher litter depth resulted in increasing $N_2O$ emissions at all CGs ($p < 0.05$).

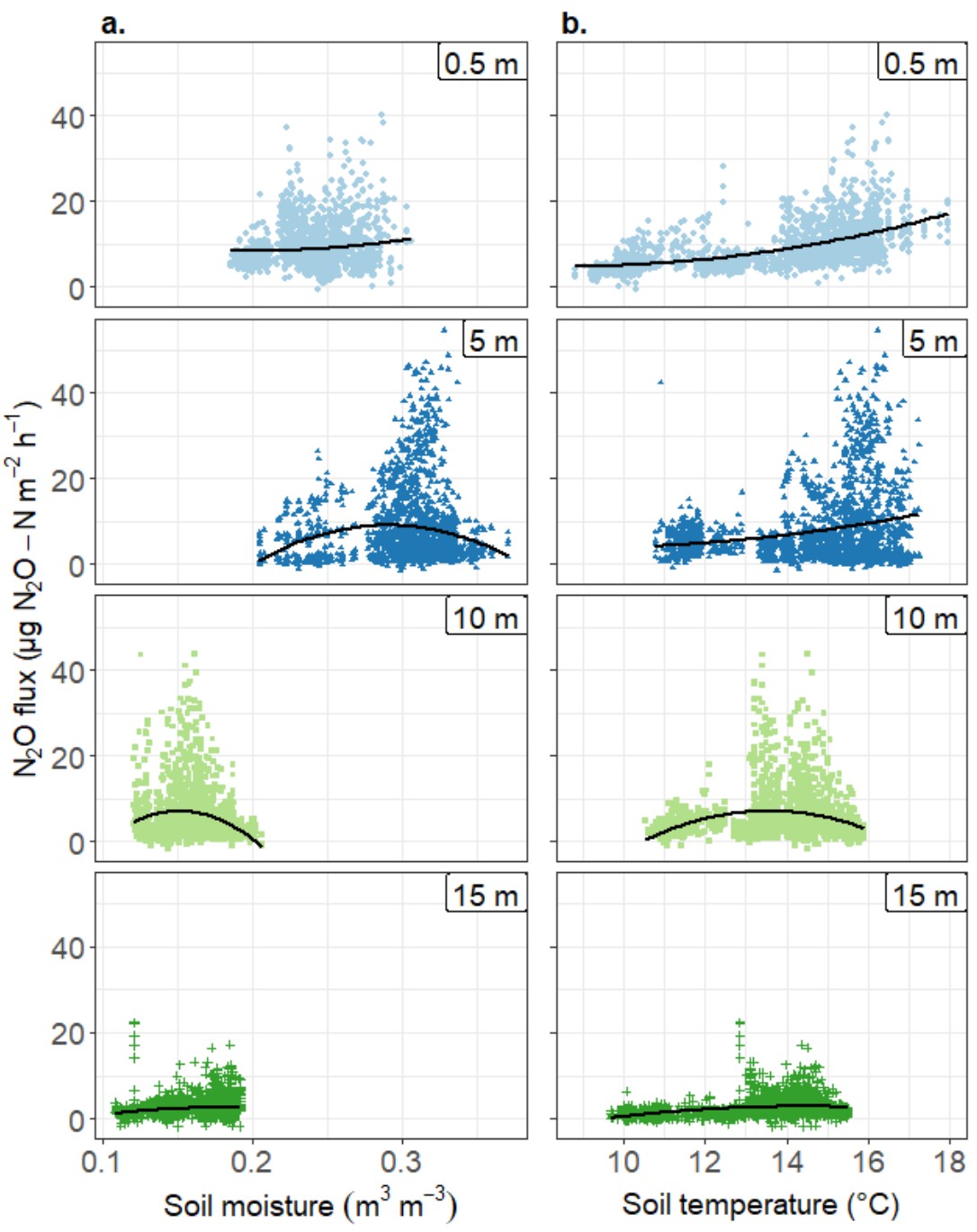


**Figure 4:** Relationship between $N_2O$ fluxes (µg $N_2$O-N $m^{-2}$ $h^{-1}$) and **a**. soil moisture ($m^3$ $m^{-3}$),
and **b**. soil temperature (°C) by distance from the stream (0.5 m, 5 m, 10 m, 15 m). Flat locations
are indicated in blue (0.5 m and 5 m) and sloped locations in green (10 m and 15 m). The fitted
lines show the linear regression on geometrically distributed data using the "geom_smooth"
function (method = "lm") from ggplot2. The $R^2$ for these regressions are shown in Table 4.

Over the 85-day measurement period, we detected episodes of $N_2O$ uptake at eleven chambers.
The measured uptake rates averaged $0.51 \pm 0.48$ µg $N_2O$-N m$^{-2}$ h$^{-1}$. $N_2O$ uptake occurred
predominantly in sloped locations (number of observations: 65 sloped, 16 flat), notably at CG15
(50 observations; Fig. 1d), and predominantly later in the measurement period (September to
November).

**Discussion**
*Soil $CO_2$ emissions*
The soil $CO_2$ emissions estimated in this study are similar to those from studies in nearby
forests, with 115.7 mg $CO_2$-C m$^{-2}$ h$^{-1}$ and 113.0 mg $CO_2$-C m$^{-2}$ h$^{-1}$ emitted in Rosalia (Leitner
et al., 2016) and at Schottenwald, near Vienna, respectively (Hahn et al., 2000). The values we
measured are only slightly lower than the average soil $CO_2$ emission from 18 different forest
ecosystems amongst Europe (Janssens et al., 2001). However, other studies in comparable
beech and spruce stands in France (Epron et al., 1999) and Germany have found values up to
50% lower (Luo et al., 2012). Apart from differences in measurement methods and seasons, it
is very likely that most of the differences can be explained by variations in soil moisture (e.g.,
Hanson et al., 1993) and temperature (e.g., Lloyd and Taylor, 1994), as discussed in the
following section.

*Effect of inclination and distance to a stream on soil $CO_2$ emissions*
Model results showed a significant negative effect of inclination, with lower soil $CO_2$ emissions
on sloped locations. This is contrary to our first hypothesis and to the findings of studies from
temperate and boreal forests in North America (Creed et al., 2013; Warner et al., 2018), where
soil $CO_2$ emissions were highest in sloped locations compared to ridge and flat locations, while
a subtropical forest in Puerto Rico showed only a weak relation between $CO_2$ fluxes and
topographic variation (Quebbeman et al., 2022). However, our results suggest that higher $CO_2$
emissions at flat locations were mainly driven by CG5, where we observed the highest $CO_2$
emissions. Being at the foot hill of the slope, CG5 likely received a large water and nutrient
input from the steep slope as compared to the other distances and had optimal conditions for
soil microbial activity. A soil texture favourable to microbial activity (enough clay to retain
moisture and enough sand to allow sufficient volatile substrate and $O_2$ access) could lead to
such a peak, but the clay content was not significantly different between CG0.5, CG5, and
CG15 nor was the sand significantly different at any distance. The effect of soil moisture on
$CO_2$ emissions was different across the CGs: at CG10, where we recorded the second-highest
emissions, soil moisture was as low as at CG15. It is possible that the high porosity at CG10
enabled an easier diffusion of $CO_2$ from the soil matrix to the atmosphere. However, even
though we found highest emissions at the wettest CG, our overall results showed higher $CO_2$
emissions with decreasing soil moisture, probably due to the negative correlation between soil
moisture and soil temperature. Indeed, the strong interaction between soil moisture and
temperature, seen in the model results for all three gases, restricts our ability to draw firm
conclusions for these variables individually. Consistent over all CGs, we found that $CO_2$
emissions increased with increasing soil temperature, in agreement with findings from, e.g.,
temperate Norway spruce and beech forests in Europe (Epron et al., 1999; Hahn et al., 2000;
Buchmann, 2000; Luo et al., 2012), where most temporal variations in the soil $CO_2$ flux could
be explained by soil temperature. The spatial variability of soil moisture and soil temperature
itself may be an effect of a different slope, its exposition and the direction from where the rain
comes. This influences the amount of rain reaching the soil surface and the evapotranspiration
of the forest, which results in a differing water balance. Compared to sites in North America
(Creed et al., 2013; Warner et al., 2018) and Germany (Buchmann, 2000), and considering the
exposition of the slope (Finke 2022, personal communication), our site is likely drier.
We suggest that the effect of inclination and distance to the stream were closely interacting with
indirect effects on soil properties and resulted in different soil $CO_2$ emissions than we expected,
notably at CG5. For example, $CO_2$ emissions were significantly lower at CG0.5 than all other
CGs, and soil pH was the highest at this distance, probably due to the close proximity to the
forest stream with a higher pH value or root-mediated changes in the pH (Hinsinger et al., 2003;
Fürst et al., 2021). Higher soil pH ($> 5$) can increase soil $CO_2$ fluxes by stimulating autotrophic
respiration from living roots and heterotrophic respiration from soil microorganisms (Reth et
al., 2005; Aciego Pietri and Brookes, 2008). However, our model results suggest increasing
$CO_2$ emissions with low soil pH values. We suggest that this is due to the chemistry in the soil,
namely the dominating carbonate species (Finke 2022, personal communication). At a low soil
pH, carbonic acid ($H_2CO_3$) dominates over carbonate ($CO_3^{2-}$), and carbonic acid might release
$CO_2$. At high pH, carbonate dominates, which can hinder $CO_2$ emissions. We encourage
researchers to analyse their sites covering a wider range of microbial communities, roots, and
soil nutrients, which might give further insight on whether soil pH directly or indirectly
influences soil $CO_2$ emissions on a topological and moisture gradient. Overall, inclination likely
had an indirect effect on the $CO_2$ emissions at our study site through its influence on soil
moisture and soil properties at the base of the slope (GC5) where the highest emissions were
measured.

*$CH_4$ uptake*
The soil $CH_4$ uptake at our site was considerably higher than values reported from other studies
in the same forest (Leitner et al., 2016), in forests near Vienna (Hahn et al., 2000), and in
Germany (Born et al., 1990; Brumme and Borken, 1999). These differing values support the
findings in forest ecosystems across Northern Europe, where temperate forest soils showed $CH_4$
uptake rates with a widely varying range between 1-165 µg $CH_4$-C $m^{-2}$ $h^{-1}$ (Smith et al., 2000).
The uptake on our sloped locations (126.9 $\pm$ 51.3 µg $CH_4$-C $m^{-2}$ $h^{-1}$) falls on the upper end of
this range. Different measurement methods, involving the use of manual chambers and gas
chromatography in nearby plots (see Leitner et al., 2016) compared to automated chambers and
laser-based gas analysers in our study, could explain the dissimilar values obtained in the same
forest ecosystem. In addition, the measurement period of this study did not cover the entire
year, which may give rise to the differences between this study and previous studies conducted
at the same site. As for soil $CO_2$ emissions, spatial variability resulting from the exposition of
the slope, and the differences in soil moisture and soil temperature, might be other reason for
our high values. Because the soils at our site are relatively dry, this might have favoured the
uptake of soil $CH_4$.

*Effect of inclination and distance to a stream on soil $CH_4$ uptake*
Opposite to our second hypothesis, soil $CH_4$ uptake was not significantly correlated with
inclination. This is opposite to the findings of other studies that did find an inclination effect.
However, the studies are not in agreement as to where uptake is higher: in a subtropical forest
in Puerto Rico, higher $CH_4$ uptake on ridges was found as compared to in valleys (Quebbeman
et al., 2022); in a temperate forest in Maryland, USA, $CH_4$ uptake was higher in transition zones
than uplands, and valley bottoms were occasionally large net sources (Warner et al., 2018); and
in a tropical forest in China, hillslopes were found to be hotspots for $CH_4$ uptake, while the
slope foot and groundwater discharge zone contributed less (Yu et al., 2021). Nonetheless, soil
$CH_4$ uptake was significantly higher at CG15 compared to CG0.5, suggesting that the distance
to the stream did have an effect on $CH_4$ uptake; the two other distances were potentially not far
enough from the stream for them to have a significant effect on the soil moisture, soil
temperature, and soil parameters that would lead to an effect on the $CH_4$ uptake. With
significant positive correlations between both litter weight and soil C content with $CH_4$ uptake,
we suggest that soil C content and litter regulated $CH_4$ uptake over distance. In agreement with
our findings, Warner et al. (2018) found a higher $CH_4$ uptake on locations with high C content
in a temperate forest landscape in Maryland, USA. Litter can hinder water from precipitation
to easily enter into the soil (Walkiewicz et al., 2021). Since there was more litter on sloped than
on flat locations, the litter could have stored the rainfall water, thus keeping the mineral soils
underneath drier at sloped locations, as has been reported in other studies (Borken and Beese,
2006; Wang et al., 2013). We therefore suggest that inclination modulated the soil $CH_4$ uptake
through its influence on weight and depth of the litter layer, and that inclination *per se* was not
the main driver of $CH_4$ uptake at our site. Instead, the weight and depth of the litter layer and
the soil C content had the largest effect on the $CH_4$ uptake.
In our study, both models showed higher $CH_4$ uptake rates with increasing soil moisture and
decreasing soil temperature. This does not only contradict findings from other forests (e.g.,
Adamsen and King, 1993; Castro et al., 1995) but cannot be distinguished visually (Fig. 3). It
is possible that our models produced ambiguous results for soil moisture and temperature,
because they were unavoidably associated in our studied *in situ* system; both variables are
influenced by inclination and distance to a stream concurrently and this thus limits our ability
to draw firm conclusions about either variable separately. Running a LMM with one variable
or the other did not help resolve this ambiguity. A long-term study in a German forest, also
found that soil moisture and soil temperature only weakly correlated with $CH_4$ uptake and were
not able to find a suitable empirical model for $CH_4$ (Luo et al., 2012). The lack of clear
relationships between soil moisture and soil temperature with $CH_4$ uptake confirms that litter
and soil C content were the best predictors of $CH_4$ uptake at our site.

*Soil $N_2O$ fluxes*
The soil $N_2O$ emissions from our site were very similar to the rates reported 200 m further
upslope from this study (Leitner et al., 2016) and in deciduous forests near Vienna (Pilegaard
et al., 2006), with values between 5.4 and 6.4 $\mu$g $N_2O$-N $m^{-2}$ $h^{-1}$, respectively. They are also
comparable to the average $N_2O$ emissions from soils in seven European coniferous forests
(Pilegaard et al., 2006), but lower than $N_2O$ emission estimates in forests subjected to high N
deposition rates in Europe (Hahn et al., 2000; Luo et al., 2012; Gundersen et al., 2012),
suggesting that N deposition was not a significant driver for the $N_2O$ emissions at our study
site. In addition to data on low $N_2O$ emissions, we provide a new dataset from a temperate
upland forest soil with reliable $N_2O$ uptake measurements, highlighting the possibility of upland
forest soils acting as $N_2O$ sink (Wrage et al., 2004; Savage et al., 2014). With the GasFluxTrailer
being a robust, state-of-the-art instrument and a total of 7670 $N_2O$ flux observations, 81
observations indicating uptake, we are confident that the $N_2O$ uptake we measured is not
instrumental noise (see Cowan et al., 2014).

*Effect of inclination and distance to a stream on soil $N_2O$ emissions*
In agreement with our third hypothesis, $N_2O$ emissions were significantly lower in sloped
locations with lower soil moisture content, which was also found by other forest soil studies in
France (Vilain et al., 2012), Kenya (Arias-Navarro et al., 2017), Australia (Butterbach-Bahl et
al., 2004), and Ecuador (Lamprea Pineda et al., 2021); although, this is opposite to the findings
in forests in China (Yu et al., 2021) and in Puerto Rico (Quebbeman et al., 2022). Furthermore,
$N_2O$ emissions in flat positions increased with increasing soil temperature. Our findings
therefore could support the hypothesis that inclination influences $N_2O$ emissions from
temperate upland forest soils. However, this soil temperature effect should be interpreted with
caution considering the concurrent, significant soil moisture:soil temperature interaction, which
could influence the significance of individual effects. $N_2O$ emissions further decreased
significantly with increasing distance to the stream. The decrease of $N_2O$ emissions from CG0.5
to CG15 might also be a consequence of the higher litter depth at these distances. The quantity
and quality of the litter input has been shown to influence $N_2O$ emissions from forests (Ambus
et al., 2006; Pilegaard et al., 2006; Walkiewicz et al., 2021), especially when coniferous needle
litter is compared with deciduous leaf litter. Moreover, tree species have been found to exert a
strong control on N cycling in forests (Lovett et al., 2004). We suggest that the thick, mostly
deciduous leaf litter layer provided a physical barrier that hindered rainfall water to easily reach
the soil matrix and thus affected $N_2O$ emissions indirectly by reducing soil moisture, which is
in line with what we suggested for the $CH_4$ uptake. Our conclusions, however, are not consistent
with a study conducted at another site in Rosalia, where removal of litter led to lowered $N_2O$
emissions (Leitner et al., 2016). This site was, however, a pure mature beech stand. Because it
is unclear how much of the total soil $N_2O$ emissions resulted from the litter layer, we suggest
that further studies repeat litter removal versus control experiments to quantify the magnitude
of $N_2O$ emissions resulting from litter. We propose that for our site, a large fraction of the N
remained stored in the litter layer and was not released as $N_2O$ (Eickenscheidt and Brumme,

527      2013).


**Conclusion**
With the state-of-the-art technology used in this study, our dataset allows a detailed look at the
influence of inclination, distance to a stream, soil moisture, soil temperature, and other soil and
litter properties on soil $CO_2$, $CH_4$, and $N_2O$ fluxes in a temperate upland forest in Eastern
Austria. Our study provides evidence of the complex interactions between inclination and
distance to a stream, and the resulting small-scale changes of soil and litter properties within an
upland forest ecosystem. We suggest that soil $CO_2$ emissions were likely indirectly affected by
inclination through its influence on soil moisture and soil properties. Contrary to our
expectations, soil $CO_2$ emissions were lower in sloped locations where soil moisture content
was lower. Our study site was a large $CH_4$ sink over the whole measurement period, with higher
soil $CH_4$ uptake rates on the locations furthest away from the stream. Because inclination was
not significantly related to the uptake of $CH_4$, we suggest that it was not a direct driver of $CH_4$
uptake at our site. Instead of soil moisture, which is commonly cited as the main driver of $CH_4$
fluxes, we found that soil C content and litter depth and weight were likely the main drivers of
$CH_4$ uptake. Our study showed a clear, significant influence of inclination and distance to the
stream on soil $N_2O$ emissions from a temperate upland forest ecosystem, which was to some

extent regulated by litter depth. We showed that the impact of inclination and distance to a stream on GHG fluxes is driven by multiple direct and indirect effects, highlighting the need to consider small-scale differences as controlling factor for future GHG flux studies to improve future GHG balance predictions in forest ecosystems.

## Statements and Declarations

### Funding

This work has been financially supported by the Austrian Research Promotion Agency (FFG), project "LTER-CWN: Long-Term Ecosystem Research Infrastructure for Carbon, Water and Nitrogen" (858024), by the Austrian Climate Research Program, 12th Call (KR19AC0K17557, ''EXAFOR'') and by the European Commission (H2020 Project eLTER PLUS, Number 871128)

### Conflicts of interest/Competing interests

The authors have no conflicts of interest to declare.

### Authors' contributions

E.D.-P designed the experiment and collected and pre-processed the data.

L.G. and N.T. analysed the data, assisted by D.A, P.F, and E.D-P.

L.G. conducted the final statistics and finalised the figures.

N.T. and L.G wrote the manuscript with extensive comments from E.D.-P.

D.A, P.F, S.G. and S.Z.-B. edited the manuscript.

E.D.-P., S.Z.-B., and S.G., were responsible for infrastructure development in the LTER-CWN project.

S.Z.-B. and S.G. were responsible for funding acquisition.

**Availability of data and material**
The gas flux data and soil and litter parameter data are stored in the online repository B2Share
and can be shared upon request. The meteorological data is open access on the online
repository B2Share: http://doi.org/10.34730/f883fa7ae62648debd6e172448cfbc9b.

**Code availability**
The code can be provided upon request.

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
