# Peer review of "Land inclination controls CO2 and N2O fluxes, but not CH4 uptake, from a"

_EGUsphere, 2023_

## Referee Comment (RC2)

**Dear Authors,**

I would like to present my evaluation of the manuscript entitled "Inclination controls CO2 and N2O fluxes, but not CH4 uptake, from a temperate upland forest soil".

This study shows the effect of slope (of the land) and distance (to the stream) on GHGs in a temperate forest soil. GHG emissions are modified by the local land conditions, slopes, and topography, and it is very important to take into account these factors when looking at landscape-scale emissions. The authors used recent technological analyzers known for their high precision and sensitivity to demonstrate how CO2, CH4, and N2O fluxes vary within a short space. However, there are a number of issues that should be addressed. The main aspects that need to be revisited are the interpretation of the results, and how Inclination and distance are regarded as factors for the changes in the emissions. Distance by itself is not a factor causing the differences in the emissions across the plots, rather the changes in the soil properties, which are of course not, modified by the distance itself. The authors may look at the historical conditions at the site. The measurements are set up close to a watercourse, but the possible floods and their consequences have not been adequately discussed in this manuscript. Frequent inundation can lead to varying soil properties, and drying-rewetting enhances decomposition. High standard deviations are highly visible in the soil properties of the site, particularly at CG5 and CG15, as presented in Table 1, which indicates high uncertainty and less confidence in the results or the number of samples. Lastly, the authors compared their findings with earlier studies in the discussion section, which is good despite incomplete year measurement in this study, but the authors should also look at other studies with similar objectives where GHG emissions are investigated with respect to the slope of the land or with reference to streams or rivers. Other specific comments are listed below:

**Title**

Please give a more specific title. The word "Inclination" can have multiple meaning. Please make it a bit clearer in the title what inclination is being referring to. Land inclination?? Slope of the land??

**Abstract**

In the keywords, why is topography included? Topography has not been discussed in this paper

**Introduction**

L44: don't need to repeat N2O as it has already been stated above in line #38

L62-63: Please revise this statement.

L91-92: The impact of topographic variation hasn't been studied so much with regard to GHG emissions. Is it due to the difficult nature of the task or the general assumption that the slope has no impact on GHG emissions?

**Methods**

L115: GasFluxTrailer is a platform. This statement sounds GasFluxTrailer was used to measure GHGs, but the trailer is the platform to position your gas analyzer.

L123: The manufacturing company name, and country is missing

L133/134: Again here. The trailer is being mentioned as a system estimating the gas exchanges. This may confuse the readers. The gas samples are analyzed by the two picarro analyzers.

L142-144: Chambers closing and opening simultaneously or successively

L200: This statement should be moved from here to the above section (Field measurements, L149-152).

**Results**

L238: According to the results in Table 1, the soils at CG0.5 and CG10 are sandier compared to the two locations, CG5 and CG15. And the clay contents of all distances are very low.

L240: Table 1: Litter depth, litter weight, soil C, porosity, organic matter, soil pH, sand content, silt content and clay content at CG5 and CG15 have very high standard deviations indicating high spatial variability and thus uncertainty. First, why such big variability have occurred within such small area? Second, why didn't you attempt to increase the number of sampling points to reduce the variability? Moreover, in none of the sections of this manuscript have I seen explanations for why these variabilities have occurred.

L254: It seems average fluxes are reported here, but cumulative fluxes are generally a better approach to compare fluxes of different treatments. Why is average flux preferred over the cumulative flux?

L273-278: There is no need to mention the significance of the main factors (soil moisture and temperature) when the interaction between the two is significant.

L282-287: R2s in Table 2 represent marginal and conditional R2 as described by the authors. However, for each regression represented in Figure 2, no R2 values are shown. The R2 and P-values should be shown in each figure.

L311-312: This is because the interaction is significant. If the interaction is significant, it is difficult to separate the variance due to the main effects.

L319-323: Figure 3: Please see the above two comments.

**Discussion**

L374-375: According to Figure 1a, the lowest CO2 emission is at CG0.5 followed by CG15, which is on the sloped location. Therefore, this statement is not true. The CO2 emission at the flat area is not significantly different from the CG15 and also the major differences between two distances occur within the flat area (CG0.5 and CG5). Thus, the values presented in Figure 1 won't enable us to conclude slope as a factor influencing the CO2 emission while the most significant difference is observed within the flat locations. Distance can also not be a factor affecting the CO2 emission.

L380-381: CG5 receiving water from the steep slope cannot favour microbial activity by itself. Is the water carrying nutrients and organic matter? Then, this might lead to changes in the microbial activity. The authors haven't said anything about the water coming from the stream. The plots are located very close to the stream and there is a high possibility that there is an interaction between the stream water and the nearby plots.

L412-414: This statement contradicts to the model results mention in L410, where decreasing CO2 was associated with low pH value.

L417-419: The results showed the main drivers of the CO2 emissions are neither the slope nor the distance from the stream. All measured results showed high spatial variability with no particular pattern to slopes or distances of the plots.

L428-431: These differences may also arise from the differences in annual climate conditions such has temperature and precipitation. Please keep in mind that this study hasn't completed the full year measurements, which may give rise to the differences between this and previous studies conducted at the same sites. This needs to be explored.

L438: Is it really distance that has an effect on CH4 uptake? Based on table 3, distances of 5 m and 10 m are not significant, even though 15 m shows significance. Soil moisture and temperature seem to be the major factor controlling the CH4 uptake rate.

L448-450: In L296, it is mentioned that CH4 is marginally affected by inclination by referring to Table 3. However, inclination is mentioned here a a non-driver of CH4 uptake. Please be consistent when the results are interpreted.

L451-452: High CH4 uptake was associated with decreasing soil moisture rather than increasing?

L453-456: The model generates what has been given to it. If the data is valid and a correct procedure is followed, the model will produce the right output. Being able to correctly interpret the model result is also critical. Interpreting the main effects separately while the interaction is significant may lead to a wrong conclusion.

L458-460: Please see the comment above.

---

## Author Comment (AC1)

Dear editorial support team,

We are pleased that the two referees were quite positive about our study, and we are grateful for the time and consideration the SOIL EGUsphere team and reviewers have put towards reviewing our manuscript. The reviewer comments were very appreciated and will help improve its quality. We have done our best to address the comments constructively, as described in more details below.

On behalf of me and my colleagues

Yours sincerely, Dr. Lauren M. Gillespie

**Response to the comments from RC1:**

**Response**: thank you very much for the nice words and constructive comments. We have addressed the minor comments individually below:

RC1: 'Comment on egusphere-2023-255', Anonymous Referee #1, 18 Apr 2023

In their manuscript the authors present results of a study using innovative monitoring technique for analyzing greenhouse gas (GHG) fluxes from forest soils in a mountainous region in Austria. The authors hypothesize that particularly inclination of slopes and the distance from an adjacent stream are important site properties which could explain differences of GHG emission or uptake. Both parameters can be expected to be closely correlated with soil properties like moisture and temperature, which both influence soil biological activity. The results of this well-planned study were carefully statistically analyzed and some of the assumptions could be confirmed, others had to be rejected. In general, the manuscript is written in a precise manner and conclusions are drawn clearly based on the presented and carefully interpreted results. The topic fits well to the scope of the journal and only some minor issues, as listed below, should be considered before the paper seems to be ready for publication.

1. 1. 50: Could you please add some information about the processes behind  $N_2O$  uptake by soils?

**Response**: We propose to add:**

"Net  $N_2O$  uptake (from the atmosphere into the soil) is a complex process closely tied to  $N_2O$  consumption (within the soil) that is driven principally by denitrifying bacteria (Liu et al. 2022)."

Although we do not have space to fully explain soil  $N_2O$  uptake and  $N_2O$  consumption in this study, since it was not the primary focus, the readers will now be able to find additional explanations in the newly added reference.

Liu, H., Li, Y., Pan, B. *et al.* Pathways of soil N2O uptake, consumption, and its driving factors: a review. *Environ Sci Pollut Res* **29**, 30850–30864 (2022). https://doi.org/10.1007/s11356-022-18619-y

2. 1. 98-102: Hypotheses:

a) In the hypotheses you speak about "inclination *per se*". Could you please explain, what is exactly meant with "*per se*"? For me inclination is mainly a proxy for other directly influencing site or soil parameters like soil moisture of temperature.

**Response**: We agree with Reviewer 1 that the three "*per se*" 's in the hypotheses do not add any meaningful information and lead to confusion. We will remove them.

b) Hypothesis No. 3 addresses potential impact on "N2O emissions". In the following results and discussion sections the term "N2O fluxes" is primarily used instead of "N2O emission".

**Response**: We thank Reviewer 1 for bringing this to our attention. We will make sure to use 'emissions' and 'uptake' when appropriate and only use 'fluxes' when referring to both emissions and uptake.

c) In the results section data on  $N_2O$  uptake are provided but there is no related hypothesis.

- **Response**: N2O uptake had only recently been measured reliably thanks to advancement in equipment and is generally minimal in comparison with emissions. We believe it important to report on this N2O uptake since its existence is interesting, but it does not merit a dedicated hypothesis since its occurrence is infrequent and unlikely to play a dominant role in soil greenhouse gas fluxes.
- 3. 116-117: Some additional information about the investigated forest sites would be helpful here. Are all locations covered by the same tree species or what about the exposition of the slopes?

**Response**: We agree and will add the exposition of the slope and the adjacent tree species.

- 4. 172/l. 237/Tab. 1: It is mentioned that the investigated soils have a larger stone content, which is also confirmed by the data in Tab. 1. However, bulk density explicitly including the coarse soil fraction with values clearly below 1.0 g cm-3 is very low and values of 0.15 or 0.12 g cm-3 seem to be unrealistic. The reason for this low bulk density should be explained and data checked (see comment on Tab. 1 below).
- **Response**: Thanks for spotting this; there was indeed an issue with the values in Table 1 (see our response to Reviewer 1's comment 6; the corrected Table 1 can be found in the Supplement document) and the 0.15 and 0.12 g cm-3 were incorrect. This has been corrected in the table at the end of our comments to the reviewers. We consider bulk densities below 1.0 g cm-3 (0.6-0.8 in our case) are realistic and common for forest soils (Beguin et al. 2017; Llek et al. 2017), even considering the stone content (7-13 % vol).

Beguin, J., Fuglstad, G. A., Mansuy, N., & Paré, D. (2017). Predicting soil properties in the Canadian boreal forest with limited data: Comparison of spatial and non-spatial statistical approaches. *Geoderma*, *306*, 195-205. Ilek, A., Kucza, J., & Szostek, M. (2017). The effect of the bulk density and the

decomposition index of organic matter on the water storage capacity of the surface layers of forest soils. *Geoderma*, 285, 27-34.

5. 218: Please write the term abbreviated as "AIC" in full when the abbreviation is used for the first time.

**Response**: Ok, this will be added.

- 6. 234-239/Tab. 1: Please check this section and the data provided in Tab. 1: According to Tab. 1 litter depth and weight has lowest values at GC5 not at GC0.5. The same is true for soil C and N content. The data shown in Tab. 1 obviously need some corrections: E.g., pH values of 0.65 or 0.34 in soils (as shown for GC5 and GC15, respectively) are extremely unlikely and also C:N ratios of 1.35 or even 0.81 are not really realistic. Most probably, some of the average values and values of standard error have been exchanged in single columns and for some of the parameters.
- **Response**: We are very thankful that Reviewer 1 brought this to our attention. There was indeed an issue with the updated table. This has now been corrected which can be found in the Supplement document. The values are now consistent with the observations made in the text.

| Variable                 | Unit              | Distance                     |                       |                           |                            |
|--------------------------|-------------------|------------------------------|-----------------------|---------------------------|----------------------------|
|                          |                   | 0.5 m                        | 5 m                   | 10 m                      | 15 m                       |
| Litter depth             | cm                | $4.4\pm0.7^{\rm a}$          | $7.0 \pm 1.2^{ab}$    | $8.5\pm1.0^{\text{b}}$    | $8.0\pm1.4^{b}$            |
| Litter weight            | g m -2 | $147.7\pm23.1^{\rm a}$       | $311.8\pm47.0^{ab}$   | $358.5\pm100.0^{ab}$      | $622.2\pm362.1^{\text{b}}$ |
| Soil N content           | %                 | $0.25\pm0.06^{\rm a}$        | $0.39\pm0.09^{ab}$    | $0.6\pm0.26^{\text{b}}$   | $0.42\pm0.18^{ab}$         |
| Soil C content           | %                 | $4.12\pm0.78^{\rm a}$        | $6.35 \pm 1.65^{ab}$  | $10.15\pm4.8^{b}$         | $7.85 \pm 4.29^{ab}$       |
| Soil CN ratio            |                   | $16.56\pm1.35^{\rm a}$       | $16.24\pm0.81^{a}$    | $17.07 \pm 1.81^{a}$      | $18.23 \pm 1.99^{a}$       |
| Bulk density*            | g cm 3 | $0.81\pm0.15^{\rm a}$        | $0.73\pm0.12^{\rm a}$ | $0.6\pm0.11^{\rm a}$      | $0.81\pm0.08^{\rm a}$      |
| Volumetric stone content | %                 | $7.59\pm8.4^{\rm a}$         | $7.84\pm2.57^{\rm a}$ | $10.79\pm2.78^{\rm a}$    | $13.16\pm2.24^{\rm a}$     |
| Porosity†                |                   | $0.75\pm0.01^{\rm a}$        | $0.79\pm0.03^{ab}$    | $0.87\pm0.04^{\rm b}$     | $0.80\pm0.02^{ab}$         |
| Organic material         | %                 | $9.25 \pm 1.4^{\rm a}$       | $13.87\pm3.73^{ab}$   | $20.86\pm8.01^{\text{b}}$ | $16.70\pm7.02^{ab}$        |
| Soil pH                  |                   | $5.57\pm0.65^{\rm a}$        | $4.00\pm0.34^{ab}$    | $4.01\pm0.34^{ab}$        | $3.78\pm0.31^{\text{b}}$   |
| Sand content             | %                 | $598.970\pm7.5^{\mathrm{a}}$ | $52.0\pm9.5^{\rm a}$  | $40.6\pm3.7^{\rm a}$      | $41.6\pm4.4^{\rm a}$       |
| Silt content             | %                 | $38.5\pm7.7^{\rm a}$         | $45.1\pm8.5^{\rm a}$  | $53.1\pm4.5^{\rm a}$      | $52.0\pm5.0^{\rm a}$       |
| Clay content             | %                 | $2.5\pm0.3^{\rm a}$          | $2.9 \pm 1.4^{\rm a}$ | $6.3\pm1.4^{\text{b}}$    | $6.5\pm0.7^{ab}$           |

**Table 1:** Average value and standard error of litter and soil parameters at each distance from the stream. "CG" indicates chamber group, with the numbers 0.5, 5, 10, and 15 defining the distance to the stream (m). Different letters indicate differences between distances (Dunn multiple comparison test after Kruskal–Wallis test, p < 0.05) for

\*with coarse material

†without coarse material

each variable.

---

## Author Comment (AC2)

Dear editorial support team,

We are pleased that the two referees were quite positive about our study, and we are grateful for the time and consideration the SOIL EGUsphere team and reviewers have put towards reviewing our manuscript. The reviewer comments were very appreciated and will help improve its quality. We have done our best to address the comments constructively, as described in more details below.

On behalf of me and my colleagues

Yours sincerely,
Dr. Lauren M. Gillespie

**Response to the comments from RC2:**

**Response:** We thank the reviewer for their comments and the concerns raised. We have, among other changes, reformulated the interpretation of results, included comments on the uncertainty, and addressed the issue with the large variability. Our study does have limitations as we don not cover the long-term dynamics of the fluxes and there are uncertainties on the interactions between several soil processes. However, we consider our high temporal resolution dataset to be robust and from a well-designed experiment that it is able to disentangle the effect land inclination on the soil GHG fluxes at this site.

Dear Authors,

I would like to present my evaluation of the manuscript entitled "Inclination controls CO2 and N2O fluxes, but not CH4 uptake, from a temperate upland forest soil".

This study shows the effect of slope (of the land) and distance (to the stream) on GHGs in a temperate forest soil. GHG emissions are modified by the local land conditions, slopes, and topography, and it is very important to take into account these factors when looking at landscape-scale emissions. The authors used recent technological analyzers known for their high precision and sensitivity to demonstrate how CO2, CH4, and N2O fluxes vary within a short space. However, there are a number of issues that should be addressed. The main aspects that need to be revisited are the interpretation of the results, and how Inclination and distance are regarded as factors for the changes in the emissions. Distance by itself is not a factor causing the differences in the emissions across the plots, rather the changes in the soil properties, which are of course not, modified by the distance itself.

**Response**: In this study, we argue that inclination and distance to a water source can be used as proxies for the variations in soil moisture, soil temperature, and other soil properties (Line 66-77). Distance to the stream and inclination influence soil water retention, which also influences soil temperature, as well as litter retention (e.g. litter descending from sloped locations and accumulating at the flat locations) and nutrient leaching.

The authors may look at the historical conditions at the site. The measurements are set up close to a watercourse, but the possible floods and their consequences have not been adequately discussed in this manuscript. Frequent inundation can lead to varying soil properties, and drying-rewetting enhances decomposition.

**Response**: We can add further detail on how the flat locations by the stream are at higher risk to be influenced by flooding and how that may have influenced sol properties. We can also add that there were no inundations or significant drying/rewetting events during the measurements.

High standard deviations are highly visible in the soil properties of the site, particularly at CG5 and CG15, as presented in Table 1, which indicates high uncertainty and less confidence in the results or the number of samples.

**Response**: There was an issue with the values in Table 1, which has now been corrected and can be found in the Supplement document. In this corrected table the standard deviations are much lower.

Lastly, the authors compared their findings with earlier studies in the discussion section, which is good despite incomplete year measurement in this study, but the authors should also look at other studies with similar objectives where GHG emissions are investigated with respect to the slope of the land or with reference to streams or rivers.

**Response**: Although we touched on this in the introduction, it is true we could better develop this in the discussion section. We will compare our findings to similar studies investigating these aspects such as:
- Arias-Navarro et al., 2017, Geophys. Res. Biogeosciences
- Davidson et al., 2000, Bioscience
- Hiltbrunner et al., 2012, Glob. Chang. Biol.
- Lamprea Pineda et al., 2021, Biogeosciences
- Quebbeman et al. 2022, Ecosystems
- Warner et al., 2018, Biogeochemistry
- Yu et al., 2008, Glob. Chang. Biol.
- Yu et al., 2021, Sci. Total Environ.

Other specific comments are listed below:
Title
Please give a more specific title. The word "Inclination" can have multiple meaning. Please make it a bit clearer in the title what inclination is being referring to. Land inclination?? Slope of the land??

**Response**: We will use "Land inclination…" in the title to remove ambiguity.

Abstract
In the keywords, why is topography included? Topography has not been discussed in this paper

**Response**: We thank Reviewer 2 for catching this, it will be replaced with 'slope inclination'.

Introduction
L44: don't need to repeat N2O as it has already been stated above in line #38

**Response**: We thank Reviewer 2 for pointing this out. It will be removed.

L62-63: Please revise this statement.

**Response**: We will replace "saturated due to $O_2$ limitation" by "water saturated".

L91-92: The impact of topographic variation hasn't been studied so much with regard to GHG emissions. Is it due to the difficult nature of the task or the general assumption that the slope has no impact on GHG emissions?

**Response**: Indeed, it has not been studied much in the past, and we believe that to a large extent this may be due to the challenging logistics required for it, in terms of equipment that captures short-term changes in gaseous emissions, appropriate chambers for the task, and a suitable site. Our state-of-the-art instrumentation and a comprehensively monitored site allowed us to overcome these challenges and provide, as the reviewer indicates, one of the few available datasets documenting the impact of topographic variation on GHG emissions in a forest soil.

Methods
L115: GasFluxTrailer is a platform. This statement sounds GasFluxTrailer was used to measure GHGs, but the trailer is the platform to position your gas analyzer.

**Response**: We refer to the GasFluxTrailer as the combination of the gas chambers, the multiplexer, and the gas analysers, as a unit. It is thanks to this unit that the GHG fluxes are measured.

L123: The manufacturing company name, and country is missing

**Response**: Thank you for pointing this out. This part should read: "to measure this gradient, one Em50 (METER Group, Inc. Pullman, WA; USA) was connected to four ECH2O 5 TM volumetric water content and temperature sensors (METER Group). One sensor was installed per CG approximately one meter away from the outer chamber (see Fig S.1)"

L133/134: Again here. The trailer is being mentioned as a system estimating the gas exchanges. This may confuse the readers. The gas samples are analyzed by the two picarro analyzers.

**Response**: Although the GHG quantification was conducted by the PICARRO analysers, we refer to the GasFluxTrailer as the unit needed for the overall flux estimation, since the trailer does not just contain the analysers but also all equipment required for the automated measurements, including for example the software used to control the entire sampling process, i.e. opening and closing of the chambers, transport of the gas from the chambers, etc.

L142-144: Chambers closing and opening simultaneously or successively

**Response**: The closing and opening is done successively. This information will be added to the text.

L200: This statement should be moved from here to the above section (Field measurements, L149-152).

**Response**: This will be done.

Results
L238: According to the results in Table 1, the soils at CG0.5 and CG10 are sandier compared to the two locations, CG5 and CG15. And the clay contents of all distances are very low.

**Response**: We are very thankful that Reviewer 2 brought this to our attention. There was indeed an issue with the updated table. This has now been corrected which can be found in the Supplement document. The values are now consistent with the observations made in the text.

L240: Table 1: Litter depth, litter weight, soil C, porosity, organic matter, soil pH, sand content, silt content and clay content at CG5 and CG15 have very high standard deviations indicating high spatial variability and thus uncertainty. First, why such big variability have occurred within such small area? Second, why didn't you attempt to increase the number of sampling points to reduce the variability? Moreover, in none of the sections of this manuscript have I seen explanations for why these variabilities have occurred.

**Response**: There was an issue when Table 1 values were updated. This has now been corrected and can be found in the Supplement document. In this corrected table the standard deviations and spatial variability are much lower.

L254: It seems average fluxes are reported here, but cumulative fluxes are generally a better approach to compare fluxes of different treatments. Why is average flux preferred over the cumulative flux?

**Response**: We chose to report average fluxes, oppose to cumulative fluxes, due to the data gaps, notably in August, that were not at the same moments between the three GHGs. In addition, we wish to avoid readers who skim the article to assume that cumulative values cover the entire year and comparing it with their or other studies.

L273-278: There is no need to mention the significance of the main factors (soil moisture and temperature) when the interaction between the two is significant.

**Response**: We humbly disagree, it is very possible for the main factors to not be significant while the interaction between the two is. We therefore chose to leave this information in the text.

L282-287: R2s in Table 2 represent marginal and conditional R2 as described by the authors. However, for each regression represented in Figure 2, no R2 values are shown. The R2 and P-values should be shown in each figure.

**Response**: We also consider that it would be ideal to include this information in the figures, and we wished to do so. However, the difficulty is making it evident to the reader what

the values are referring to. We were unable to find a clear way to indicate what values were from the distance model and which were from inclination model and then the p-values associated to each explanatory variable and interactions. For $CO_2$ emissions, for example, we would need to include 14 values (4 $R^2$ values and 10 p-values) plus labeling on the figure, which would make it illegible. Moreover, distributing them amongst the different panels would make it very difficult to understand what they were referring to, and the reader would be dependent on the value labels. Although not ideal, we believe it is much easier for the reader to indicate in the figure legend to find the r2 and p-values in the tables.

L311-312: This is because the interaction is significant. If the interaction is significant, it is difficult to separate the variance due to the main effects.

**Response**: Thanks for pointing this out, this caveat is underlined in the discussion (L453-456).

L319-323: Figure 3: Please see the above two comments.

**Response**: Please see the response to the comment about Figure 2.

Discussion

L374-375: According to Figure 1a, the lowest CO2 emission is at CG0.5 followed by CG15, which is on the sloped location. Therefore, this statement is not true. The CO2 emission at the flat area is not significantly different from the CG15 and also the major differences between two distances occur within the flat area (CG0.5 and CG5). Thus, the values presented in Figure 1 won't enable us to conclude slope as a factor influencing the CO2 emission while the most significant difference is observed within the flat locations. Distance can also not be a factor affecting the CO2 emission.

**Response**: The confusion likely originates from the fact of having two positions on flat locations (CG0.5 and CG5) and two on sloped ones (CG10 and CG15). Overall, the fluxes from flat locations were significantly lower than sloped locations as indicated by the inclination model. We agree with Reviewer 2 that we can soften this statement about the influence of inclination and immediately call attention to the fact that the highest values were at the middle distances as written on line 378.
We propose:
"Model results showed a significant negative effect of inclination, with lower soil $CO_2$ emissions on sloped locations, which contrary to our first hypothesis and to the findings of studies from temperate and boreal forests in North America (Creed et al., 2013; Warner et al., 2018) where soil $CO_2$ emissions were highest in sloped locations compared to ridge and flat locations. However, our results suggest that higher CO2 emissions at flat locations were mainly driven by CG5, where we observed the highest CO2 emissions. Being at the foot of the slope…"

L380-381: CG5 receiving water from the steep slope cannot favour microbial activity by itself. Is the water carrying nutrients and organic matter? Then, this might lead to changes in the microbial activity. The authors haven't said anything about the water coming from the stream. The plots are located very close to the stream and there is a high possibility that there is an interaction between the stream water and the nearby plots.

**Response**: We disagree that water cannot stimulate microbial activity by itself; several drying and rewetting studies support the influence of water content on microbial activity. We agree thought that water accumulation could have also assisted in higher nutrient content, and we will add this to the text:
"Being at the foot hill of the slope, CG5 likely received a larger water and nutrient input from the steep slope as compared to the other distances and had optimal conditions for soil microbial activity"

L412-414: This statement contradicts to the model results mention in L410, where decreasing $CO_2$ was associated with low pH value.

**Response**: We thank Reviewer 2 for pointing this out. The statement on line L410 is incorrect, there is a negative relationship between soil $CO_2$ emissions and pH, which means higher $CO_2$ emissions are correlated with lower pH values. This will be corrected. The results are therefore consistent with the statement on L412-414.

L417-419: The results showed the main drivers of the $CO_2$ emissions are neither the slope nor the distance from the stream. All measured results showed high spatial variability with no particular pattern to slopes or distances of the plots.

**Response**: We understand Reviewer 2´s concern that this statement may be too strong for the results shown. We propose: "We conclude that inclination likely had an indirect effect on the $CO_2$ emissions at our study site through its influence on soil moisture and soil properties at the base of the slope (GC5) where the highest emissions were measured."

L428-431: These differences may also arise from the differences in annual climate conditions such has temperature and precipitation. Please keep in mind that this study hasn't completed the full year measurements, which may give rise to the differences between this and previous studies conducted at the same sites. This needs to be explored.

**Response**: We agree with Reviewer 2 that the differences in climate conditions and different time periods measured could be better underlined. We will underline this more in the text.

L438: Is it really distance that has an effect on CH4 uptake? Based on table 3, distances of 5 m and 10 m are not significant, even though 15 m shows significance. Soil moisture and temperature seem to be the major factor controlling the CH4 uptake rate.

**Response**: It is possible that the two other distances were not far enough from the stream for them to have a significant effect, and Figure S2 shows soil moisture and soil texture to be influenced by either inclination and/or distance. We agree though that we can better re-iterate the influences of distance and inclination on soil moisture and soil temperature.

L448-450: In L296, it is mentioned that CH4 is marginally affected by inclination by referring to Table 3. However, inclination is mentioned here a a non-driver of CH4 uptake. Please be consistent when the results are interpreted.

**Response**: In the results, we report on data and model results, while in the discussion we interpret these results. We do not find that the indicated sentences contradict what is stated in the results but interprets and concludes based on the ensemble of results.

L451-452: High CH4 uptake was associated with decreasing soil moisture rather than increasing?

**Response**: The model results shown in Table 3 indicate a positive correlation between $CH_4$ uptake and soil moisture. This result was certainly unexpected for us as well. As we discuss in L451-460, we believe that factors other than soil moisture may have had a strong influence on $CH_4$ uptake during our study.

L453-456: The model generates what has been given to it. If the data is valid and a correct procedure is followed, the model will produce the right output. Being able to correctly interpret the model result is also critical. Interpreting the main effects separately while the interaction is significant may lead to a wrong conclusion.
L458-460: Please see the comment above.

**Response**: Thank you for pointing out this. We believe that the point the referee is underlining is precisely what we are trying to explain in the indicated lines: it is soil moisture and temperature combined that they need to be looked up, because they are unavoidable associated. We will rewrite this sentence in the revised manuscript to clarify the message we are trying to convey.

**Table 1:** Average value and standard error of litter and soil parameters at each distance from the stream. "CG" indicates chamber group, with the numbers 0.5, 5, 10, and 15 defining the distance to the stream (m). Different letters indicate differences between distances (Dunn multiple comparison test after Kruskal–Wallis test, $p < 0.05$) for

| Variable | Unit | Distance | | | |
|---|---|---|---|---|---|
| | | 0.5 m | 5 m | 10 m | 15 m |
| Litter depth | cm | $4.4 \pm 0.7^a$ | $7.0 \pm 1.2^{ab}$ | $8.5 \pm 1.0^b$ | $8.0 \pm 1.4^b$ |
| Litter weight | g m$^{-2}$ | $147.7 \pm 23.1^a$ | $311.8 \pm 47.0^{ab}$ | $358.5 \pm 100.0^{ab}$ | $622.2 \pm 362.1^b$ |
| Soil N content | % | $0.25 \pm 0.06^a$ | $0.39 \pm 0.09^{ab}$ | $0.6 \pm 0.26^b$ | $0.42 \pm 0.18^{ab}$ |
| Soil C content | % | $4.12 \pm 0.78^a$ | $6.35 \pm 1.65^{ab}$ | $10.15 \pm 4.8^b$ | $7.85 \pm 4.29^{ab}$ |
| Soil CN ratio | | $16.56 \pm 1.35^a$ | $16.24 \pm 0.81^a$ | $17.07 \pm 1.81^a$ | $18.23 \pm 1.99^a$ |
| Bulk density* | g cm$^3$ | $0.81 \pm 0.15^a$ | $0.73 \pm 0.12^a$ | $0.6 \pm 0.11^a$ | $0.81 \pm 0.08^a$ |
| Volumetric stone content | % | $7.59 \pm 8.4^a$ | $7.84 \pm 2.57^a$ | $10.79 \pm 2.78^a$ | $13.16 \pm 2.24^a$ |
| Porosity† | | $0.75 \pm 0.01^a$ | $0.79 \pm 0.03^{ab}$ | $0.87 \pm 0.04^b$ | $0.80 \pm 0.02^{ab}$ |
| Organic material | % | $9.25 \pm 1.4^a$ | $13.87 \pm 3.73^{ab}$ | $20.86 \pm 8.01^b$ | $16.70 \pm 7.02^{ab}$ |
| Soil pH | | $5.57 \pm 0.65^a$ | $4.00 \pm 0.34^{ab}$ | $4.01 \pm 0.34^{ab}$ | $3.78 \pm 0.31^b$ |
| Sand content | % | $598.970 \pm 7.5^a$ | $52.0 \pm 9.5^a$ | $40.6 \pm 3.7^a$ | $41.6 \pm 4.4^a$ |
| Silt content | % | $38.5 \pm 7.7^a$ | $45.1 \pm 8.5^a$ | $53.1 \pm 4.5^a$ | $52.0 \pm 5.0^a$ |
| Clay content | % | $2.5 \pm 0.3^a$ | $2.9 \pm 1.4^a$ | $6.3 \pm 1.4^b$ | $6.5 \pm 0.7^{ab}$ |

*with coarse material

†without coarse material

each variable.